# Immunotherapeutic Approaches in Ovarian Cancer

Hyunho Yoon [1,2], Ayoung Kim [1] and Hoon Jang [3,*]

1   Department of Medical and Biological Sciences, The Catholic University of Korea,
    Bucheon 14662, Republic of Korea
2   Department of Biotechnology, The Catholic University of Korea, Bucheon 14662, Republic of Korea
3   Department of Life Sciences, Jeonbuk National University, Jeonju 54896, Republic of Korea
*   Correspondence: hoonj@jbnu.ac.kr; Tel.: +82-63-270-3359

**Abstract:** Ovarian cancer (OC) is gynecological cancer, and diagnosis and treatment are continuously advancing. Next-generation sequencing (NGS)-based diagnoses have emerged as novel methods for identifying molecules and pathways in cancer research. The NGS-based applications have expanded in OC research for early detection and identification of aberrant genes and dysregulation pathways, demonstrating comprehensive views of the entire transcriptome, such as fusion genes, genetic mutations, and gene expression profiling. Coinciding with advances in NGS-based diagnosis, treatment strategies for OC, such as molecular targeted therapy and immunotherapy, have also advanced. Immunotherapy is effective against many other cancers, and its efficacy against OC has also been demonstrated at the clinical phase. In this review, we describe several NGS-based applications for therapeutic targets of OC, and introduce current immunotherapeutic strategies, including vaccines, checkpoint inhibitors, and chimeric antigen receptor (CAR)-T cell transplantation, for effective diagnosis and treatment of OC.

**Keywords:** next-generation sequencing; ovarian cancer; immunotherapy





## 1. Introduction

Gynecological cancer is a type of cancer that occurs in female reproductive organs, such as the vulva, vagina, cervix, uterus, fallopian tubes, and ovaries, that is related to the secretion of sex hormones [1]. Ovarian cancer (OC) is the eighth most common cancer in women, with a 5-year survival rate of approximately 45% [2,3]. The incidence of OC is approximately 10% that of breast cancer. OC can be classified into several histological types, characterized by cancer risk factors, clinical features, and molecular expression [4], and they include high-grade serous carcinoma (HGSC), endometrioid carcinoma of the ovary (ECO), clear cell carcinoma (CCC), mucinous carcinoma (MC), and low-grade serous carcinoma (LGSC) [5]. LGSC is associated with serous borderline tumor mutations in KRAS and BRAF. In contrast, HGSCs are not associated with serous borderline tumor mutations and have TP53 and BRCA mutations [6,7]. As these OC types are distinct diseases, diagnosis or treatment requires methodological approaches, suggesting that histopathological diagnosis of reproducible tumor cell types is critical for the successful treatment of OC.

In general, OC that has progressed to the advanced stage of a malignant tumor shows intractability, relapse, and drug resistance, resulting in poor prognosis with conventional treatment. The first-line standard care of patients with OC is surgery with platinum-based agent-based chemotherapy [8]. The combination of taxane compounds, such as paclitaxel, and platinum-based agents, such as carboplatin, is mainly adopted as first-line chemotherapy in OC [9]. However, in most patients who receive first-line treatment, OC relapses within several years [10,11]. More than 25% of patients with relapse have a poor prognosis due to a platinum-resistant or platinum-refractory disease [12,13]. Although outcomes and response rates are low, platinum-resistant patients are treated with non-platinum agents, such as liposomal doxorubicin, melphalan, paclitaxel, gemcitabine, and topotecan [1,14,15].

With an increasing understanding of cancer progression mechanisms, targeted therapies are emerging as innovative and promising therapeutic strategies. Therefore, antiangiogenic molecules, immune checkpoint inhibitors, poly ADP-ribose polymerase (PARP) inhibitors, estrogen receptor inhibitors, and various inhibitors of intrinsic tumor signaling pathways are being used as potential therapeutic agents for OC patients [1].

In particular, PARP inhibitors, such as olaparib or niraparib, have been mainly used in OC patients with BRCA mutations [16–19], and new chemotherapy, that induces drug combinations, is constantly being developed. PARP inhibitors play a key role in the maintenance of therapy for recurrent OC and primary HGSC [20]. A study has demonstrated that PARP inhibitors remarkably extend progression-free survival (PFS) after response to platinum-based chemotherapy [21]. However, relapse after first-line treatment and toxicity of chemotherapeutic agents are still high, so it is necessary to find an effective and safe treatment method that reduces side effects.

Recent studies have shown that next-generation sequencing (NGS)-based approaches provide a rationale for sequence-matching therapies to improve the overall drug response and survival rates in cancer patients [22]. The genetic landscape of cancer patients, including mutational status and gene expression patterns, may provide new druggable targets and pathways, leading to a clinical advantage for cancer patients [23–25]. Thus, a variety of NGS-based applications, such as whole transcriptome sequencing (RNA-seq) and assays for transposase-accessible chromatin sequencing (scATAC-seq), have been used to investigate the detailed genetic landscape of OC. These efforts not only provide detailed information for the understanding of the genetic variation of OC but also form the basis for immunotherapy, a new therapeutic strategy for the treatment of OC.

In this review, we discuss how therapeutic targets were discovered through NGS-based techniques and outline the future direction by summarizing the findings of current clinical trials on the efficacy of immunotherapy in OC.

## 2. NGS-Based Identification of New Pathways and Targets in OC

### 2.1. Bulk RNA-Sequencing Analysis (RNA-Seq)

New approaches for the treatment of OC are being intensively developed. In OC patients with BRCA1/2 mutations, PARP inhibitors improved drug response rates and prognosis by preventing the repair of damaged DNA, offering a different mechanism of anticancer treatment from conventional therapy, and evaluating the status of BRCA to prevent resistance to treatments [26–28]. Also, a recent study reported that the landscape of germline alterations could enhance the diagnosis, prognosis, prediction, and strategy of therapy in OC [29]. This suggests that molecular-targeted therapy may provide an appropriate strategy for the design of new clinical trials on OC treatment. Thus, the identification of molecular markers and signaling pathways relevant to the prognosis of OC is a key step in molecular targeted therapy. Bulk RNA-seq has been widely used in transcriptome studies as a replacement for microarrays. It has an average expression level in different cell types of samples [30].

CD151 is one of the molecules that regulates the migration and invasion of OC cells, as identified by RNA-seq [31]. The global expression pattern of OC, which represents both the early and late stages of OC, was investigated using RNA-seq. Interestingly, CD151 was expressed at all stages and did not differ between stages or specific subtypes. CD151 is restricted to the cell membranes on the surface of normal ovaries, indicating that when it is internalized into the cytoplasmic endosomes, it can dissociate from the primary tumor by weakening its ability to cooperate with other cells [32]. In vivo study targeting CD151 would affect cell survival and anti-CD151 monoclonal antibodies, which, involved in metastasis, may facilitate preventing dissemination of tumor cell and peritoneal spread, which is a critical cause of lethality in HGSC [31]. Moreover, depending on its expression level, CD151 has functional significance, as it can silence or block cell migration and invasion, suggesting that it could be an important emerging therapeutic target in OC.

RNA-seq-based mutation analysis may also be useful for the detection of racial differences in genetic variants of OC. New mutations, such as TP53 (67%), BRCA1 (38%), BRCA2 (54%), and ARID1A (13%), were observed in 24 Japanese OC specimens [33]. Although >90% of HGSC cases had TP53 mutations [34], not all TP53 mutations were observed in recurrent samples, because of sample heterogeneity in the same patient. Nevertheless, the use of RNA-seq for the investigation of genes that are not transcriptionally expressed, and those that are silenced after transcription, is limited.

RNA-seq has also been used to identify long non-coding RNAs (lncRNAs), particularly long intergenic non-coding RNAs (lincRNAs), which are mainly expressed in OC tissues [35]. lincRNAs are transcribed non-coding RNAs longer than 200 nucleotides, which do not overlap annotated coding genes. They share several features with lncRNA transcripts and compose more than half of lncRNAs in humans [36]. RNA-seq analysis revealed that OC with intergenic noncoding *RNA1* (*OIN1*) is often more expressed in OC than in normal ovarian tissue. In addition, *OIN1* expression negatively correlated with the expression of apoptosis-associated genes, ras-associated domain family member 5 (*RASSF5*), and adenosine A1 receptor (*ADORA1*), which were upregulated by OIN1 knockdown in OC. Moreover, increased expression of an oncogenic gene, known as urothelial cancer-associated 1 (*UCA1*), enhanced cell migration, invasion, and cisplatin resistance in OC [37]. These results demonstrate that OIN1 may have an ameliorating effect on OC progression by modulating oncogenes, suggesting that it has the potential to play an important role in the diagnosis and treatment of OC.

However, the expression of fusion genes in OC has rarely been reported. A fusion gene is a subtype of a hybrid gene containing two or more independent genes [38]. Recently, RNA-seq approaches detected two fusion genes, *DPP9-PPP6R3* and *DPP9-PLIN3*, in advanced OC [39]. DPP9 is involved in the regulation of proliferative and survival pathways, and has the ability to suppress tumor cells [40]. In addition, the *MUC1-TRIM46-KRTCAP2* fusion gene was detected by RNA-seq in HGSC [39]. MUC1 overexpression represents a diagnostic marker in metastatic and advanced OC, and these three fusion genes are readily detected only in tumor samples, suggesting that they may be potential biomarkers for diagnostic and therapeutic targets of OC [41].

RNA-seq analysis can be used to identify important factors related to OC stem cells (CSC). ZIP4, a Zn transporter, is present in the stem cell niche and has intestinal integrity [42]. It has been reported that the loss of *ZIP4*, which is upregulated in aggressive OC, decreased some cancer stem cell-related activities, such as tumor proliferation, anoikis resistance, colony formation, and particularly, spheroid formation, suggesting that ZIP4 is a promising and significant CSC regulator in OC [43]. A mechanistic study revealed that ZIP4 promoted the activation of lipid lysophosphatic acid (LPA) in an extracellular Zn-dependent manner, acting as a ligand for the nuclear receptor peroxisome proliferator-activated receptor gamma (PPARγ) [44]. This study demonstrated that the LPA–PPARγ–ZIP4 pathway potentiated ZIP4-positive tumors by promoting CSCs in human OC, indicating that ZIP4 is an important therapeutic target.

In general, human tumor cells are heterogeneous and complex, as they require the ability to migrate to attack tissues. The PI3K/AKT/mTOR pathway activates cell proliferation and migration, and inhibits apoptosis; therefore, the PI3K pathway is a good predictor of invasive and migratory abilities of human OC [45]. High levels of AKT activation have been observed in high-grade and late-stage serous OC [46]. This study showed that AKT exhibited anti-apoptotic effects and interfered with cell cycle arrest through the mTOR pathway. One study identified five genes that were differentially expressed in OC using the RNA-seq approach, which included *ECM1*, *GPCR*, *NUR 77*, *AKT3*, and *PRKCB*; these genes are the main components of the PI3K/AKT/mTOR pathway [47]. The study implied that the activation of the PI3K/AKT/mTOR signaling pathway led to improved invasive and migratory capacities of human OC cells. Therefore, controlling this pathway could be an effective therapeutic strategy for OC.

## 2.2. Single-Cell RNA-Sequencing Analysis (scRNA-Seq)

Although bulk RNA-seq-based approaches can elucidate the mechanisms associated with cancer development, OC metastasis, and various therapeutic targets, these approaches do not allow for accurate single-cell unit analysis, because they analyze all the cells in cancer tissue [48]. Heterogeneity is also a hallmark of OC, indicating that OC cells are composed of diverse populations with distinct genotypes and phenotypes [49]. scRNA-seq approaches can determine gene expression levels for each transcript and the expression distribution in each subpopulation of cells, resulting in the precise characterization of tumor formation. scRNA-seq, through quantitative and qualitative analyses of complex tissue cells, is essential for better understanding of cellular heterogeneity in OC studies, such as those related to metastasis, drug resistance, and treatment [50].

*CXCL12* is a biomarker of OC, and its overexpression indicates the growth of OC [51]. Matrix metalloproteinases (MMPs) play a key role in cancer cell invasion and metastasis by remodeling the tumor microenvironment [52]. scRNA-seq of OC tissue has revealed that the increased expression of some collagens and *MMP* genes mostly occurs in cancer fibroblasts and stromal cells [50]. C3, a trigger in the complement system that promotes tumor development and angiogenesis, is highly expressed by metastatic fibroblasts. Moreover, surface deposition of C3 is often observed in OC patients [53]. Metastatic fibroblasts also highly express CXCL12, which plays a pivotal role in CXCR4 signaling in OC and several cancers, creating an inflammatory environment implicated in the development and growth of OC [50].

Li et al. performed scRNA-seq analysis with 66 HGSC cells to identify 6000 variable and signature genes in each cluster, sorted by principal component analysis (PCA). Differentially expressed genes are predominantly abundant in metabolic and cancer-associated pathways, including the PI3K-Akt signaling pathway, which plays an essential role in malignant cancers and is involved in tumor proliferation, survival, and angiogenesis [54]. The expression of *STAT1*, *ANP32E*, *GPRC5A*, and *EGFL6* was higher in OC tissues than in normal tissues. In addition, the expression levels of *PMP22*, *FBXO21*, and *CYB5R3* were significantly lower in OC tissues than in normal tissues. High expression of *ANP32E*, *STAT1*, *GPRC5A*, *EGFL6*, and *PMP22* implies low survival in patients with OC, whereas low expression of *FBXO21*, *ANP32E*, and *CYB5R3* indicates a longer recurrence-free survival period in OC [55]. These genes are closely associated with progression, metastasis, and cancer stem cells in various solid cancers. For example, *STAT1* and *ANP32E* are overexpressed in OC, and this leads to a worse prognosis. In addition, *GPRC5A* mutation is closely related to the induction of self-renewal in bladder cancer. Taken together, these reports demonstrate that the marker genes *ANP32E*, *STAT1*, *GPRC5A*, *EGFL6*, *PMP22*, *FBXO21*, and *CYB5R3* are strongly associated with OC prognosis [55].

## 2.3. Transposase-Accessible Chromatin Analysis by Sequencing (ATAC-Seq)

ATAC-seq is a method for mapping chromatin accessibility to DNA sequences using Tn5 transposase at the genomic level. It is widely used to acquire open chromatin and transcription-factor-binding regions for the identification of specific chromatin accessibility conditions [56]. Methodologically, a transposase carrying a known DNA sequence tag must be attached to the nucleus, followed by the opening of the chromatin at the insertion site [57]. Open chromatin is then recognized as a label of a known sequence, which is used to construct a library for sequencing [58].

In cancer research, ATAC-seq can detect the tissue-specific chromatin activity of regulatory regions in tumors. Previous studies have shown that mutant cohesins can enhance the chromatin accessibility of binding sites for transcription factors, including *ERG*, *GATA2*, and *RUNX1* [59]. In addition, the protein encoded by the well-known cancer suppressor gene p53 suppresses cancer in the normal state but promotes tumor development in the mutant state [60]. ATAC-seq was used to confirm the pre-binding of p53 to enhancers and promoters in normal fibroblasts. When DNA is damaged, inaccessible chromatin becomes accessible and p53 is activated to maintain stability [59].

Gallon et al. identified enhancers associated with platinum resistance and found *SOX9* to be an important cluster of cis-regulatory elements that regulate transcription factors [61]. *SOX9* also plays an important role in the chemical resistance of OC cells [62]. In addition, patients with OC treated with platinum chemotherapy, with a high expression of *THBS1*, a gene that is downregulated in resistant cell lines, showed longer survival [61]. Denser chromatin regions have more adduct formation during resistance, particularly in the intergenic region, indicating that more platinum-induced damage occurs in the clusters of cis-regulatory elements in chromatin regions that develop into denser nucleosomes as drug resistance progresses. Therefore, targeting chromatin structures in a specific genomic context could be an important strategy in epigenetic therapy to overcome drug resistance.

Tissue-resident macrophages (TRMs) are a heterogeneous population of immune cells that perform tissue-specific and niche-specific functions [63]. They are present in all tissues and participate in homeostasis, immunity, and tissue repair. In OC cells, ATAC-seq revealed that macrophages residing in serous cavity tissues were significantly enriched in retinoic X receptor (RXR) transcripts and RXR response elements [64]. It has also been shown that the peritoneal serous cavity TRMs (LPM) infiltrate early ovarian tumors and that RXR deletion reduces LPM accumulation in tumors, particularly in mice [64]. This suggests that the RXR signaling pathway regulates the maintenance of serous macrophages and that peritoneal LPM may be a good target for OC therapy.

### 2.4. Combined Sequencing Analysis

Advances in NGS-based applications using OC samples have elucidated the genetic landscape of patient tumors. Moreover, additional integrative analysis of NGS-based applications has provided further insight into tumor heterogeneity and detailed genetic information [65]. The role of the E3 ubiquitin ligase RNF20 and histone H2B monoubiquitination (H2Bub1), at the level of whole-genome changes in chromatin status, was investigated by gene expression analysis using ATAC-seq and RNA-seq [66]. The findings showed that in cells where RNF20 was deleted, the ATAC-seq read counts were enriched in the promoter region. Loss of H2Bub1 may lead to a more open chromatin structure in mammalian cells, implying a direct relationship between the open chromatin structure and loss of H2Bub1 [66]. In cells where RNF20 has been deleted, RNA-seq and ATAC-seq data showed a total of 1086 overlapping genes. The loss of H2Bub1, which is associated with immune pathways, can significantly alter IL-6-mediated immune signaling genes [66]. Moreover, the loss of H2Bub1 makes chromatin more accessible, suggesting that chromatin induces the upregulation of immunomodulators that promote the growth and migration of early HGSC states [66] (Figure 1 and Table 1).

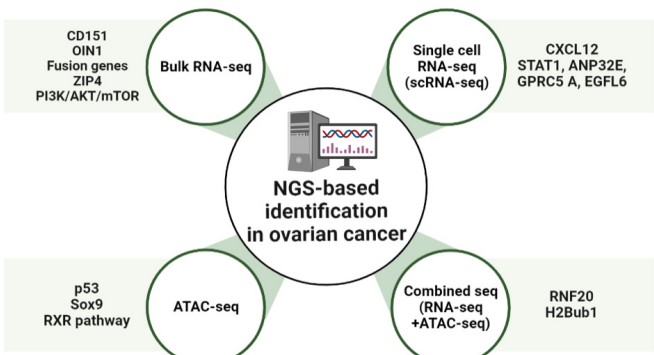

**Figure 1.** Overview of NGS-based identification in ovarian cancer (OC). Schematic showing molecular targets of OC treatment. OIN1, Ovarian cancer long intergenic noncoding RNA 1; CXCL12, C-X-C motif chemokine 12; ANP32E, Acidic Nuclear Phosphoprotein 32 Family Member E; GPRC5A, G-protein coupled receptor family C group 5 member A; EGFL6, Epidermal growth factor-like domain-containing protein 6; RXR, Retinoid X receptor; RNF20, Ring finger protein 20.

**Table 1.** Molecular targets discovered through NGS in ovarian cancer.

| Seq. Type | Molecule | Function | Reference |
|---|---|---|---|
| Bulk RNA-seq | OIN1(ovarian cancer long intergenic noncoding RNA1) | Unannotated lincRNA originated from chromosome 10q21; Contribute to ovarian cancer progression and lead to suppression of apoptosis | [37] |
| | CD151 | Maintaining normal cellular integrity, cell-to-cell communication, wound healing, platelet aggregation, trafficking, cell motility and angiogenesis; Association with various stages of cancer, including metastatic cascade and primary tumor growth | [67,68] |
| | UCA1 (urothelial cancer-associated 1) | Biomarker for diagnosis of bladder cancer; Promote proliferation, migration, and immune response in gastric cancer | [69] |
| | MUC1 | Single pass type I transmembrane protein with a heavily glycosylated extracellular domain (oncoprotein); Protection to underlying epithelia; Maintain pluripotency and self-renewal ability in embryonic stem cells | [70] |
| | ZIP4 | A zinc transporter; Related to the process of tumor growth and metastasis of many cancers | [71] |
| | PI3K–AKT–mTOR pathway | A group of plasma membrane-associated lipid kinases; Regulate a broad spectrum of cellular mechanisms involving survival, proliferation, growth, metabolism, angiogenesis and metastasis | [72,73] |
| scRNA-seq | CXCL12 | Ligand for CXCR4; Important factor in physiological and pathological processes involving embryogenesis, hematopoiesis, angiogenesis and inflammation | [74] |
| | ANP32E | Interact with a short region of the docking domain of H2A.Z chaperon | [75,76] |
| | STAT1 | Regulate a variety of cellular processes, such as antimicrobial activities, cell proliferation and cell death | [75,76] |
| ATAC-seq | TP53 (p53) | Tumor suppressor gene; Inhibit cell division or survival in response to various stresses | [77] |
| | SOX9 | Transcription factor; Determine cell fate during embryonic development | [78] |
| | RXR (Retinoid X receptor) | Regulate various functions by virtue of their dimerization with other nuclear hormone receptors, leading to activities of different signaling pathways | [79] |
| Combined seq | H2Bub1 | weaken DNA–histone interactions and interrupt chromatin compaction | [80] |

## 3. Immunotherapy in OC

In cancer research, immunotherapy enhances the immune response such that the patient's immune cells can kill cancer cells. It is effective against several tumors and has innovative therapeutic potential to replace chemotherapy as the next-generation treatment for gynecological cancers [81]. Immunotherapy can be categorized into active and passive types. Active immunotherapy directly targets specific cancer antigens, and this involves treatment with immunostimulatory vaccines, genetically modified to recognize target antigens or chimeric antigen receptor T cells (CAR-T). Passive immunotherapy involves treatment with cytokines or checkpoint inhibitors, which indirectly activate a patient's immune response [82]. These treatments are innovative, but according to the results of clinical trials, only some cancers can be treated, and side effects, such as cytokine syndrome and decreased immune function, can occur [81,83].

### 3.1. Immune-Stimulating Peptide Vaccines

Vaccine-mediated immunostimulatory therapy involves the use of a specific antigenic vaccine of the tumor, which is an immunotherapeutic agent that suppresses tumor growth or cancer recurrence by inducing an immune response [84,85]. Peptide vaccines consist of DNA vectors expressing dendritic cells, recognized by autologous tumor-specific antigens or tumor-specific peptides that induce immune activation, and the activity of which can be enhanced by various immune modulators [86,87].

### 3.1.1. P53 Peptide Vaccine

The most common mutant gene in human cancers is p53, with nearly 50% of OCs having p53 mutations [88]. Because p53 mutations mainly increase the expression and stability of the p53 protein, the amount of p53 in cancer cells is higher than that in normal cells [89]. Therefore, a p53 peptide vaccine could be a targeted therapeutic strategy for various cancer cells. It has been reported that p53-synthetic long peptide (p53-SLP) vaccine treatment

in OC patients significantly induced p53-SLP-specific T-cell responses [90]. In addition, low-dose cyclophosphamide pre-treatment in patients with recurrent OC improved the immunogenicity of the p53-SLP vaccine, by enhancing the activity of regulatory T cells [91]. In a clinical study on platinum-resistant OC, p53 vaccine and gemcitabine significantly increased the reactivity of T cells, leading to better survival [92]. Recently, studies on the combination of various immunomodulators and p53 vaccines have been conducted to improve biochemical sensitivity and immune responses [93,94].

### 3.1.2. Wilms' Tumor 1 (WT1) Peptide Vaccine

WT1 is often detected in various cancers, including leukemia, lung cancer, breast cancer, thyroid cancer, melanoma cancer, and OC [95]. Mouse and human *WT1* genes have high homology, and hence, a murine model was used for WT1 vaccine development [95]. After confirming that the WT1 vaccine inhibited the growth of cancer cells by inducing a strong immune response, a WT1 vaccine using serum from cancer patients was evaluated for the first time [95]. HLA-A*24:02-restricted WT1-targeting vaccine promoted peptide-specific humoral immunity in refractory OC patients; therefore, WT1 levels can be used as a potential prognostic marker [96].

### 3.1.3. New York Esophageal Squamous Cell Carcinoma-1 (NYESO-1) Peptide Vaccine

NYESO-1 is a marker expressed in various cancer cells, including OC, and it is a significant target for vaccines [97]. Vaccine-mediated CD4+ and CD8+ T-cell responses, and the application of NY-ESO-1+ lymphocytes, have been reported [98]. The modified NY-ESO-1 vaccine rapidly induced a consistent and safe immune response in most vaccinated OC patients [99]. A recent study reported that the NYESO1 vaccine linked to the secretin-penetratin peptide induced a stronger and more specific T-cell immune response in a mouse animal model [100].

### 3.1.4. Glypican-3 (GPC3)-Derived Peptide Vaccine

GPC3, a heparan sulfate proteoglycan bound to the cell membrane by glycosylphosphatidyli-nositol, is specifically expressed in liver cancer, OC, lung cancer, and melanoma [101,102]. A recent report showed that the GPC3 vaccine improved the survival rate of chemoresistant OCs after chemotherapy [103]. In addition, it has been reported that adjuvant miRNA enhances the effect of GPC3 vaccine, and that the combination of miR-375, 193a, and 1128 can be used as predictive biomarkers [104]. The GPC3 vaccine was used in combination with CAR-T therapy, to significantly enhance the therapeutic effect against OC, suggesting that the GPC3 vaccine could be used in combination with several cancer treatments in the future [105].

### 3.1.5. Dendritic Cell (DC)-Based Peptide Vaccine

DCs are antigen-presenting cells of the immune system and are widely recognized as important cell types that can initiate antitumor responses [106]. DC-based vaccines are a novel strategy for effectively treating cancer patients, and hence, various clinical studies have been conducted using DC-related responses in cancer patients. Correll et al. reported that a DC vaccine could inhibit the activity of regulatory T cells in the blood of melanoma patients, and subsequently restore T-cell activity [107]. As a result of these observations, DC vaccines were evaluated in several carcinomas. Autologous DC-based vaccines with tumor lysates after chemotherapy in OC patients have also been reported to successfully reduce the rate of cancer progression and improve survival [108]. In addition, DC vaccines related to neoantigen peptides are being developed as clinical candidates for immune enhancement [109]. DC-based vaccines have great potential to improve the treatment of OC patients, and the advantage of using the immune system to induce a more sustained response, compared with that during cytotoxic chemotherapy [110]. However, further clinical studies of a cohort of patients with OC using DC vaccines are needed to confirm its safety and efficacy.

### 3.2. Blockade of Checkpoint

Checkpoint inhibitors are widely used to treat refractory tumors, including OC. Immune checkpoints are generally regulated by negative feedback inhibition, and they protect the host from autoimmunity and maintain self-tolerance by regulating the responses of various immune cells [111,112]. When activated T-cells recognize and bind to specific markers of tumor cells, immune checkpoints rapidly regulate the immune response through the T-cell receptor (TCR) signaling pathway [113].

### 3.2.1. Cytotoxic T-lymphocyte Antigen 4 (CTLA-4) Inhibitor

CTLA-4 inhibitors are widely used as immune checkpoint blockers to induce an immune response. PHI-101, a checkpoint kinase 2 inhibitor, showed a tumor-reducing effect on platinum-resistant, recurrent OC [114]. PHI-101 therapy was assessed in a phase IA clinical trial, and a phase II clinical trial has been recommended for the same. Ipilimumab, under the trade name Yervoy, is a CTLA-4 antibody generally used in the treatment of various cancers [115]. However, the anti-CTLA-4 antibodies need improvement, because immunotoxicity has often been reported in the liver, gastrointestinal tract, and endocrine system within the first few weeks of treatment [116].

### 3.2.2. Programed Cell Death Protein 1 (PD-1) and Programed Cell Death-Ligand 1 (PD-L1) Inhibitors

PD-1, a co-inhibitory receptor, inhibits T-cell activation, suggesting that it is a target for immunotherapy in cancer cells. PD-1 has two types of potential ligands: PD-L1 and PD-L2. PD-L1 is mainly expressed in most hematopoietic cells and vascular endothelial cells, whereas PD-L2 is expressed in some macrophages and dendritic cells [117]. PD-L1 is generally found in various malignancies, including OC [118,119]. PD-1–PD-L1 interaction normally inhibits the survival, proliferation, and functions of T cells [120]. Blocking the interaction of PD-1 with PD-L1 is widely used clinically for the treatment of tumors, and this strategy is less toxic than the use of CTLA-4 inhibitors, such as ipilimumab [121].

Pembrolizumab, also known as Keytruda, is a PD-1 inhibitor, and it has been effective in patients with recurrent OC [122,123]. A recent study showed that the combination of pembrolizumab and PEGylated liposomal doxorubicin has a potential therapeutic effect on platinum-resistant OC [124]. The combination of pembrolizumab with bevacizumab dramatically decreased the serum CA-125 level and regression of recurrent masses, with no marked side effects [125].

Nivolumab, under the trade name Opdivo, is a human monoclonal antibody approved for the treatment of several malignancies; it enhanced the anticancer activity of T cells by blocking the PD-1 and PD-L1/L2 pathways [118,126,127]. Monotherapy with nivolumab exhibited low tumor specificity and response in platinum-resistant OC [126]. Co-administration of nivolumab with various adjuvants has shown optimistic trends in the treatment of various cancers, including OC. In addition, the combination of nivolumab with the anti-CTLA4 antibody, ipilimumab, has been reported to improve the overall response to OC [128].

Atezolizumab (tecentriq; MPDL3280A) is an immunoglobulin mAb, that selectively interacts with PD-L1 in cancer cells, in the tumor microenvironment, and reactivates suppressed T cells to kill malignant tumors [129]. As PD-L1 expression is highly detectable in OC specimens, atezolizumab has been attracting attention as a potential immunotherapeutic agent. In addition, a combination of atezolizumab and other immunotherapeutic agents showed preliminary clinical activity in patients with OC [130]. For example, the combination of atezolizumab and bevacizumab induces a sustained response in some patients with platinum-resistant OC [131].

Abelumab, a monoclonal antibody capable of mediating antibody-dependent cytotoxicity, is a checkpoint inhibitor that interacts with PD-L1, especially when administered in combination with chemotherapeutic agents [132,133]. Several clinical studies have shown that avelumab monotherapy is effective in Merkel cell carcinoma (MCC) and urothelial

cancer [134]. In addition, it has been reported that combination therapy with avelumab and axitinib increased the survival rate of urothelial cancer patients and showed significant activity when combined with docetaxel [134]. Several combination strategies have recently been evaluated, and further studies are underway to improve the response rate of avelumab in patients with OC [135–137].

### 3.3. Chimeric Antigen Receptor T (CAR-T) Cells

CAR-T cells are genetically modified patient-derived immune cells designed to activate the immune response by recognizing specific surface antigens on cancer cells. The inhibition of histone deacetylase (HDAC) activity plays an important role in CAR-T cell immune recognition. Co-treatment with sodium valproate (VPA), a representative HDAC inhibitor, and CAR-T enhanced the immune recognition of CAR-T cells in OC [138]. The identification of specific antigens overexpressed in cancer cells is an important strategy in CAR-T therapy. The most common target antigens for CAR-T in OC are mesothelin, mucin 16 (MUC16), folate receptor $\alpha$, and human epidermal growth factor receptor 2 (HER2) [139].

Mesothelin is a new antigen that can be a target of CAR-T and is overexpressed in various cancers, including OC. However, it is also expressed in some normal tissues and has the disadvantage of causing off-target effects [140]. Mouse studies have shown that treatment with mesothelin-induced CAR-T cells is a potential therapy for OC. It was able to significantly prolong the survival of mice with OC [141].

Mucine 16 (MUC16), a glycoprotein of the mucin family, is expressed in various tumor cells and is involved in the proliferation and metastasis of cancer cells [142]. MUC16 is strongly expressed in most OCs and known well as a tumor marker (CA125), because it is cleaved and released from peripheral blood, suggesting that MUC16 is an ideal antigenic target for CAR-T. It has been reported that PD1-anti-MUC16 CAR-T cells have more potent anticancer activity than single MUC16-CAR-T cells in OC animal models [143,144]. The clinical application of MUC16 is still lacking due to several limitations, but immunotherapeutic studies using CAR-T cell construction are ongoing [145].

Folate receptor-$\alpha$ (FR$\alpha$) protein is expressed at low levels in normal cells, specifically in OC [146]. T-cell activation using CAR targeting with FR$\alpha$ has been evaluated for use in OC treatment [147]. Recently, CAR-modified, cytokine-induced killer cells with FR$\alpha$, enhanced anticancer immunity against OC [148].

Human epidermal growth factor receptor 2 (HER-2) is overexpressed in breast cancer and OC [149]. One study reported that radiolabeled pertuzumab for HER2 imaging enables rapid and unambiguous delineation of OCs overexpressing HER2 [150]. The suppression of HER2 using shHER2-RNA treatment with cisplatin also enhanced the anticancer effect of OC [151]. The study of HER2-interact synthetic Notch CAR cells has been investigated in a mouse model, and it is expected that clinical therapeutics for HER2-CAR-T cells will also be developed in the near future [152] (Table 2 and Figure 2).

**Table 2.** Immunotherapies for OC.

| Type of Mechanism | Agents | FDA Approval | Combination with | Ref. |
|---|---|---|---|---|
| Peptide vaccines | | | | |
| P53 | P53-SLP vaccine | no | Cyclophosphamide; Gemcitabine | [90–92] |
| WT1 | WT1 vaccine | no | | [95,96] |
| NYESO-1 | NYESO-1 vaccine | no | Secretin-penetratin miR-375, miR-93a, miR-1128; CAR-T cell | [97–100] |
| GPC3 | GPC3 vaccine | no | | [103–105] |
| DC | DC protein vaccine | yes | | [107] |
| DC | Neoantigen vaccine | no | | [108] |

**Table 2.** *Cont.*

| Type of Mechanism | Agents | FDA Approval | Combination with | Ref. |
|---|---|---|---|---|
| Checkpoint inhibitor | | | | |
| CTLA-4 | PHI-101 | yes | | [114] |
| CTLA-4 | Ipilimumab | yes | | [115] |
| PD-1 | Pembrolizumab | yes | Doxorubicin; Bevacizumab | [122–125] |
| PD-1 | Nivolumab | yes | Ipilimumab | [126,128] |
| PD-L1 | Atezolizumab | yes | Bevacizumab | [130,131] |
| PD-L1 | Avelumab | yes | Axitinib; Docetaxel; Doxolubicin | [134–136] |
| PD-L1 | Avelumab | yes | Carboplatin + Paclitaxel | [137] |
| CAR-T | | | | |
| Mesothelin | MSTL-CAR-T | no | | [141] |
| MUC16 | PD1-MUC16-CAR-T | no | | [144] |
| FRα | FRα-CAR-T | no | | [147] |
| FRα | FRα-CAR-T | no | Cytokine-inducing killer cell | [148] |
| HER2 | HER2-CAR-T | no | Pertuzumab | [150] |
| HER2 | HER2-CAR-T | no | Synthetic Notch | [152] |

p53-SLP, p53-synthetic long peptide; WT1, Wilms' tumor 1; NYESO-1, New York esophageal squamous cell carcinoma-1; GPC3, Glypican-3; DC, dendritic cell; CAR-T, chimeric antigen receptor T; CTLA-4, cytotoxic T-lymphocyte antigen 4; PD-1, programmed cell death protein 1; PD-L1, programmed cell death-ligand 1; MUC16, mucin 16; FRα, folate receptor-α; HER2, human epidermal growth factor receptor 2.

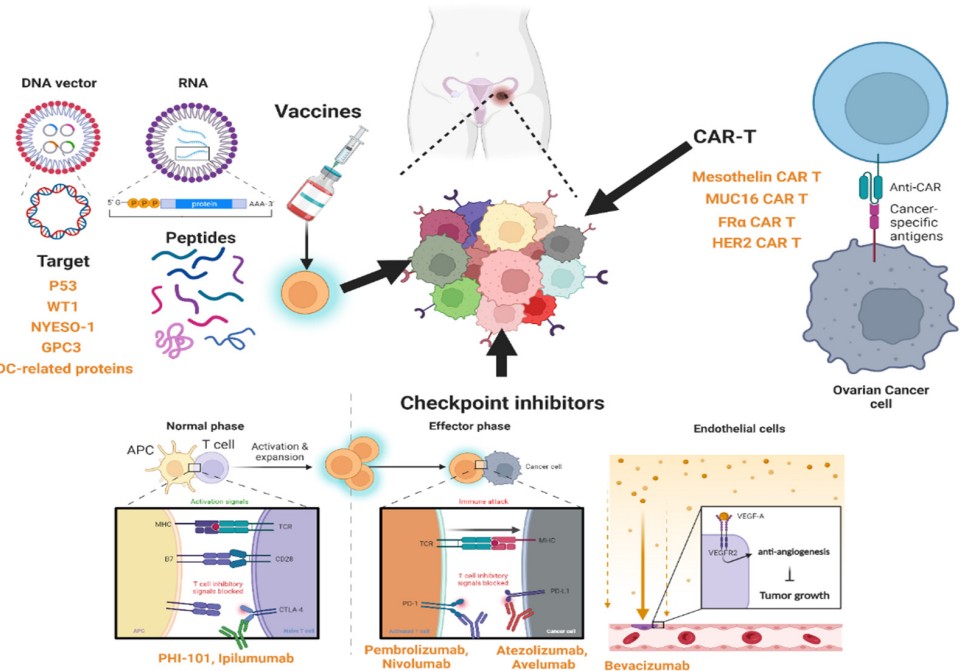

**Figure 2.** Overview of immunotherapy in OC. Schematic showing the types and mechanisms of T-cell-mediated immunotherapy currently used in clinical practice. WT1, Wilms' tumor 1; NYESO-1, New York esophageal squamous cell carcinoma-1; GPC3, Glypican-3; DC, Dendritic cell; CAR-T, Chimeric Antigen Receptor T; CTLA-4, Cytotoxic T-lymphocyte antigen 4; PD-1, Programed cell death protein 1; PD-L1, Programed cell death-ligand 1; MUC16, Mucin 16; FRα, Folate receptor-α; HER2, human epidermal growth factor receptor 2.

## 4. Conclusions

NGS-based applications have identified the detailed genetic landscape of cancer cells, including OC cells, providing different cancer treatment targets and strategies that can significantly improve patients' prognoses. However, OC remains an incurable carcinoma that threatens the life of patients due to drug resistance and recurrence. The immune system plays an important role in the pathogenesis and treatment of OC, and hence, immunotherapy to suppress cancer cells by enhancing patients' immunity has been continuously attempted in OC. The main directions of therapeutic approaches for OC reported so far are immune and vaccine-based treatment, checkpoint inhibition, and genetically engineered chimeric cells. Despite promising research results, the widespread clinical application of immunotherapy for OC is limited by insufficient experimental evidence. Therefore, epidemiological and clinical studies involving a broad population of OC patients, and expanded use of immunotherapeutic agents, are essential for improved patient clinical outcomes and enhanced survival of OC patients.

**Author Contributions:** H.Y. designed and wrote the manuscript; A.K. wrote the manuscript; H.J. supervised, organized, and wrote the manuscript; all authors revised the manuscript. All authors have read and agreed to the published version of the manuscript.

**Funding:** This research was funded by Brain Korea 21 (BK21; # M2022B002600003), the Research Base Construction Fund Support Program funded by Jeonbuk National University in 2022, and the National Research Foundation of Korea (NRF), grant provided by the Korean government (MSIT) (No. 2021R1C1C1011346).

**Institutional Review Board Statement:** Not applicable.

**Informed Consent Statement:** Not applicable.

**Data Availability Statement:** Not applicable.

**Conflicts of Interest:** The authors declare no conflict of interest.

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
