# Peer review of "Immunotherapeutic Approaches in Ovarian Cancer"

_cimb, doi:10.3390/cimb45020081_

Round 1
Reviewer 1 Report
The manuscript submitted for review is a review paper on the use of NGS methods in patients with ovarian cancer. This is an important and contemporary topic.
Overall, the manuscript is well written. The first part does not bring much new information (e.g. on PARP inhibitors) because a lot of statements "maybe" "probably" and "may be a target" - it does not bring anything to the treatment of women with ovarian cancer.
On the other hand, the second part regarding the review of therapeutic methods is valuable.
My comments:
1. Many mistakes in the introduction:
a. The ovaries are not just reproductive organs (l25)
b. Ovarian cancer is not the third cancer (l27)
c. Ovarian cancer does not cause more deaths than breast cancer (l29)
d. meaning "serous borders" (l34)
e. meaning "abnormal BRCA function" (l35)
2. International names of drugs are supplemented by the statement "known as". Does this apply to trade names? (l320, l340)
3. What do the authors mean by the statement "immunotoxicity .. needs to be improved" (l321)
With minor corrections, the manuscript is ready for publication as a review.
Author Response
Thank you for your review and feedback on our manuscript, “Next Generation Sequencing-Based Approach for Ovarian Cancer: Molecular Targeted Therapy and Immunotherapy” for consideration in Current Issues in Molecular Biology. As suggested by the reviewer, we modified the title to 'Immunotherapeutic Approaches in Ovarian Cancer'.
We appreciate the insightful comments and the opportunity to provide a revised manuscript. We have responded to the reviewer’s concerns individually. We feel confident that all of the reviewer's concerns have been addressed and that the overall quality of the manuscript has been improved. We send a track-changed version of the manuscript to the reviewers.
Thank you for your time and consideration. 
The manuscript submitted for review is a review paper on the use of NGS methods in patients with ovarian cancer. This is an important and contemporary topic.
Overall, the manuscript is well written. The first part does not bring much new information (e.g. on PARP inhibitors) because a lot of statements "maybe" "probably" and "may be a target" - it does not bring anything to the treatment of women with ovarian cancer.
On the other hand, the second part regarding the review of therapeutic methods is valuable.
Response: We appreciate the reviewer’s summary and valuavle comments on our manuscript. We totally agree with the reviewer’s critique. First, as suggested by the reviewer, a detailed description of PARP inhibitor treatment was added to the introduction. Also, we respond with point-by-point answers to each comment, including additional details.
- Many mistakes in the introduction:
- The ovaries are not just reproductive organs (l25)
Response: We agree with it. The sentences were a little ambiguous, so we modified it to ‘Gynecological cancer is a type of cancer that occurs in female reproductive organs, such as the vulva, vagina, cervix, uterus, fallopian tubes, and ovaries which are related to the secretion of sex hormones’.
- Ovarian cancer is not the third cancer (l27)
Response: Thanks for the point. Ovarian cancer is the eighth most common cancer in women. We have changed it.
- Ovarian cancer does not cause more deaths than breast cancer (l29)
Response: You are right. We included that sentence by mistake. So we removed the sentence.
- meaning "serous borders" (l34)
Response: This means serous borderline tumors. For clarity, we have rephrased it.
- meaning "abnormal BRCA function" (l35)
Response: It means that ‘Mutant BRCA works abnormally in ovarian cancer’. For clarity, we have replaced “abnormal BRCA function” with “BRCA mutation” in the sentence.
- International names of drugs are supplemented by the statement "known as". Does this apply to trade names? (l320, l340)
Response: No, it does not apply to trade names. For clarity, we have replaced it with “Ipilimumab, under the trade name Yervoy; Nivolumab, under the trade name Opdivo” in the sentence.
- What do the authors mean by the statement "immunotoxicity .. needs to be improved" (l321)
Response: This means that the anti-CTLA-4 antibodies need improvement because immunotoxicity has often been reported in the liver, gastrointestinal tract, and endocrine system within the first few weeks of treatment. For clarity, we have replaced it with the sentence above.

Reviewer 2 Report
The manuscript by Hyunho Yoon and colleagues is a review about the novel biomarkers in ovarian cancer (OC) and the associated therapies.
The review is interesting and well-written, however there are some major revisions to address.
The main problem of the manuscript is the structure (and the title). In chapter 2, the authors divide the biomarkers by NGS technique, focusing mainly on RNA-seq. In chapter 3, they list the types of immunotherapy in OC (and this part has little to do with NGS).
I suggest focusing on the biomarkers and the associated therapies more than on the techniques (also modifying the title which is misleading).
There are also some minor revisions that need to be performed.
According to the GLOBOCAN estimates, OC is the eighth cancer in females (not the third).
I would enrich the part of the other targeted therapies in OC, such as the use of PARP inhibitors.
The authors do not treat the molecular alterations found in the germline DNA of patients with OC, which have, however, an important role in the treatment (e.g. see Gurioli et al. IJMS 2022).
In general, I would like to suggest moving the references to the end of the paragraph not inside it or in the middle of the sentence.
Many paragraphs lack references, but it’s necessary to add them so the readers can find easily more information (e.g. 138-147, 175-181, 246-251, 262-264, 309-314, 321-323 etc.)
Line 46 -47: In addition to the treatments of OC also Melphalan and Paclitaxel can be involved (e.g. see PMID: 34367999 and PMID: 12440808).
Line 64 -66: All biomarkers described are effective therapeutic targets?
Line 69 -72: In addition to target therapies the evaluation of BRCA status is useful also to prevent resistance mechanisms to treatments (e.g. PMID: 35626108 and PMID: 34360649).
Can you add a table or a column to table 2, to clarify if there are clinical trials of official FDA approval for these drugs in ovarian cancer, it will help in immunotherapies comprehension.
Line 412: Table 2 and Figure 2 should be mentioned not here (in the conclusions) but before, in the Immunotherapy chapter.
Table 1: The authors write “target” but they do not specify the drugs that target these molecules.
All the biomarkers described in the text should be distinguished into diagnostic, predictive or prognostic markers.
Line 340: Nivolumab, known as “Opdivo” (not Opidivo).
Line 90: 90% of HGSC is very high, is it referred to the previously cited study?
Line 95: Which lincRNA do you mean? Can you specify?
Line 78-86: The authors described mainly in vitro studies, I would add some references about the translational relevance of CD151 evaluation.
Finally, check the entire manuscript for errors and typos (gene names must be in italics, acronyms cited at first appearance etc.)
Author Response
Thank you for your review and feedback on our manuscript, “Next Generation Sequencing-Based Approach for Ovarian Cancer: Molecular Targeted Therapy and Immunotherapy” for consideration in Current Issues in Molecular Biology. As suggested by the reviewer, we modified the title to 'Immunotherapeutic Approaches in Ovarian Cancer'.
We appreciate the insightful comments and the opportunity to provide a revised manuscript. We have responded to the reviewer’s concerns individually. We feel confident that all of the reviewer's concerns have been addressed and that the overall quality of the manuscript has been improved. We send a track-changed version of the manuscript to the reviewers.
Thank you for your time and consideration. 
Reviewer #2 (Comments to the Author):
The manuscript by Hyunho Yoon and colleagues is a review about the novel biomarkers in ovarian cancer (OC) and the associated therapies.
The review is interesting and well-written, however there are some major revisions to address.
The main problem of the manuscript is the structure (and the title). In chapter 2, the authors divide the biomarkers by NGS technique, focusing mainly on RNA-seq. In chapter 3, they list the types of immunotherapy in OC (and this part has little to do with NGS). I suggest focusing on the biomarkers and the associated therapies more than on the techniques (also modifying the title which is misleading).
Response: We appreciate the reviewer’s summary and valuavle comments on our manuscript. We agree with the reviewer’s critique. Therefore, we changed the title and phased the abstract to show that we are focusing more on biomarkers and related therapies for ovarian cancer.
There are also some minor revisions that need to be performed.
- According to the GLOBOCAN estimates, OC is the eighth cancer in females (not the third).
Response: Thanks for the point. We included that sentence by mistake. Ovarian cancer is the eighth most common cancer in women. We have changed it.
- I would enrich the part of the other targeted therapies in OC, such as the use of PARP inhibitors.
Response: We agree with the suggestion. As suggested by the reviewer, a detailed description of PARP inhibitor treatment was added to the introduction.
- The authors do not treat the molecular alterations found in the germline DNA of patients with OC, which have, however, an important role in the treatment (e.g. see Gurioli et al. IJMS 2022).
Response: We agree with the reviewer’s comments. We added the information and cite the reference in the manuscript.
- In general, I would like to suggest moving the references to the end of the paragraph not inside it or in the middle of the sentence.
Response: We agree. So, we've moved all references to the end of paragraphs.
- Many paragraphs lack references, but it’s necessary to add them so the readers can find easily more information (e.g. 138-147, 175-181, 246-251, 262-264, 309-314, 321-323 etc.)
Response: We agree. We have added more references to give more information.
- Line 46 -47: In addition to the treatments of OC also Melphalan and Paclitaxel can be involved (e.g. see PMID: 34367999 and PMID: 12440808).
Response: We added the references in the text.
- Line 64 -66: All biomarkers described are effective therapeutic targets?
Response: No. The sentence is slightly exaggerated, so we modified it in the text.
- Line 69 -72: In addition to target therapies the evaluation of BRCA status is useful also to prevent resistance mechanisms to treatments (e.g. PMID: 35626108 and PMID: 34360649).
Response: Thanks for the information. We added the information and cited the references in the manuscript.
- Can you add a table or a column to table 2, to clarify if there are clinical trials of official FDA approval for these drugs in ovarian cancer, it will help in immunotherapies comprehension.
Response: That is a good idea. As the reviewer’s suggestion, we added the information to the table 2.
- Line 412: Table 2 and Figure 2 should be mentioned not here (in the conclusions) but before, in the Immunotherapy chapter.
Response: We agree. We moved them into the immunotherapy chapter.
- Table 1: The authors write “target” but they do not specify the drugs that target these molecules.
Response: That is good point. They are molecules that may or may not be targets by drugs so, we modified term “molecule”.
- Table 1: All the biomarkers described in the text should be distinguished into diagnostic, predictive or prognostic markers.
Response: We agree that all biomarkers should be classified as diagnostic, predictive or prognostic markers, such as UCA1, a biomarker for bladder cancer diagnosis. However, not all molecules described in the manuscript act as biomarkers in cancer. Clearly, the molecules described above could be promising biomarkers, but further research is still needed. Therefore, only some biomarkers currently used in clinical practice are shown in the table.
- Line 340: Nivolumab, known as “Opdivo” (not Opidivo).
Response: Thanks for the point. For clarity, we have replaced it with “Nivolumab, under the trade name Opdivo” in the sentence.
- Line 90: 90% of HGSC is very high, is it referred to the previously cited study?
Response: Previous studies have reported that HGSC is the most common subtype of OC and TP53 mutations are almost 100% of cases. We additionally cited that study in line 98 of manuscript.
- Line 95: Which lincRNA do you mean? Can you specify?
Response: We have spicified more information about lincRNA in the text. “lincRNAs are transcribed non-coding RNAs longer than 200 nucleotides which do not overlap annotated coding genes. They share several features with lncRNA transcripts and composes more than half of lncRNA in human”.
- Line 78-86: The authors described mainly in vitro studies, I would add some references about the translational relevance of CD151 evaluation.
Response: We agree. We have added some information and references regarding in vivo studies of CD151.
- Finally, check the entire manuscript for errors and typos (gene names must be in italics, acronyms cited at first appearance etc.)
Response: We agree. We have corrected some typos and changed some gene names to italics.

Round 2
Reviewer 2 Report
I do not have further questions for the authors.